# State of the Art of Probiotic Use in Neonatal Intensive Care Units in French-Speaking European Countries

**DOI:** 10.3390/children10121889

**Published:** 2023-12-05

**Authors:** Amélie Blanchetière, Charles Dolladille, Isabelle Goyer, Olivier Join-Lambert, Laura Fazilleau

**Affiliations:** 1Department of Neonatology, University Hospital of Caen, 14000 Caen, France; 2Pharmaco-Epidemiology Unit, Department of Cardiology, University Hospital of Caen, 14000 Caen, France; 3Department of Pharmacy, University Hospital of Caen, 14000 Caen, France; 4Department of Microbiology, University Hospital of Caen, 14000 Caen, France

**Keywords:** probiotics, necrotizing enterocolitis, neonatal sepsis, neonatal morbidity, newborn, preterm infant, neonatal intensive care unit, microbiota

## Abstract

The effectiveness of probiotics in reducing the incidence of necrotizing enterocolitis has been supported by a very large number of studies. However, the utilization of probiotics in preterm infants remains a topic of debate. This study aims to assess the rate of probiotic use in European neonatal intensive care units (NICUs), compare administration protocols, and identify barriers and concerns associated with probiotic use. An online questionnaire was distributed via email to European NICUs between October 2020 and June 2021. Different questions related to the frequency of probiotic use were proposed. Data on probiotic administration protocols and reasons for non-utilization were collected. The majority of responses were from France and Switzerland, with response rates of 85% and 89%, respectively. A total of 21% of French NICUs and 100% of Swiss NICUs reported routine probiotic use. There was significant heterogeneity in probiotic administration protocols, including variations in probiotic strains, administration, and treatment duration. The main obstacles to routine probiotic use were the absence of recommendations, lack of consensus on strain selection, insufficient scientific evidence, and concerns regarding potential adverse effects. The rate of routine probiotic administration remains low in European NICUs, with heterogeneity among protocols. Further trials are necessary to elucidate optimal treatment modalities and ensure safety of administration.

## 1. Introduction

Probiotics are living microorganisms that, when administered in adequate amounts, confer a health benefit on the host [1] They encompass various taxa, including yeasts (*Saccharomyces* spp.…) and bacteria (*Lactobacillus* spp., *Bifidobacterium* spp. …), which can be used in various clinical situations such as the prevention or treatment of antibiotic-associated diarrhea, gastroenteritis and functional bowel disorders [2]. Probiotic products come in a wide range of formulations and dosages, containing a single or multiple strains. Most probiotics are available as food supplements, while some are recognized as medications with marketing authorization from the European Medicines Agency (EMA).

Probiotics play a crucial role in regulating the digestive microbiota, using a combination of mechanisms that are still not fully understood. Well-established mechanisms include their immunomodulatory effects, such as the recruitment of lymphocytes to the intestinal mucosa, enhanced antigen transport, facilitating a more rapid immune response and simulation of the intestinal tight-junction proteins. Probiotics also exert direct antibacterial action and outcompete intestinal pathogens, thus aiding the colonization of beneficial commensal bacteria [3].

In healthy full-term infants, the colonization of the digestive tract by micro-organisms occurs gradually after birth. This process is influenced by various factors, such as the mode of delivery and feeding method [4]. But other elements are also involved: breastfeeding and absence of post-natal antibiotic therapy further contribute to a greater diversity of microbiota [5].

In premature infants, gut colonization occurs when gastrointestinal function and immune system are still immature. This immaturity of the immune system leads to a reduced level of immune factors (antibodies, intestinal mucus, antimicrobial peptides…), and increased pro-inflammatory factors such as Toll-like receptors [6]. As a result of this physiological immaturity and environmental factors associated with hospitalization, colonization occurs more slowly and with less diversity in preterm infants compared to full-term infants [7]. Fecal samples from preterm infants with necrotizing enterocolitis (NEC) often exhibit a higher abundance of *Proteobacteria* spp. than those from healthy preterm infants. This bacterial phylum is associated with intestinal dysbiosis, and the dysbiosis associated with its overabundance is known to precede the onset of NEC [8,9]. Furthermore, the composition of stool microbiota differs between healthy preterm infants and those with NEC, with the latter displaying even lower microbial diversity, and a prevalence of a single bacterial species [10].

NEC remains a poorly understood disease, with a relatively stable incidence over the last years, ranging from 2 to 7% among preterm infants under 32 gestational weeks (GW), and a mortality rate of 15 to 30% [11]. Regarding this dysbiosis in preterm infants as possible cause for prematurity complications such as NEC, the administration of probiotics in this vulnerable population holds promise for restoring microbiota balance.

Since the 1990s, several placebo-controlled studies have investigated the efficacy of probiotics to prevent NEC in preterm infants, yielding promising initial results. Large-scale trials have shown a decrease in the incidence of NEC and NEC-related mortality [12]. Subsequently, numerous randomized controlled trials have been conducted among preterm infants, continuing to provide valuable insights into the use of probiotics. The most recent systematic review, which included over 10,000 premature infants, demonstrated a significative reduction in NEC risk (RR 0.54, 95% CI 0.45–0.65) with probiotic supplementation [13].

These trials have also revealed additional benefits of probiotics in preterm infants, including a decrease in the incidence of late-onset sepsis (LOS) and overall mortality. The administration of probiotics resulted in a 12–14% reduction in LOS incidence [14,15]. Furthermore, there is evidence suggesting a 24% decrease in mortality rates, although the certainty of this evidence is currently low [13].

Probiotics have also shown positive effects on other outcomes, such as reduction in the time required to achieve full enteral feeding by 1.5 days [16], and a shorter length of hospital stay by 3.8 days (5400 newborns) [17]. Importantly, the administration of probiotics does not lead to an increased risk of adverse effects on intraventricular hemorrhage, neurodevelopment, bronchopulmonary dysplasia, periventricular leukomalacia or retinopathy of prematurity, thus suggesting an excellent safety profile [17,18,19,20].

In addition to these benefits, a pilot study revealed a decrease in the diversity of expressed antibiotic resistance genes in the gut microbiome of preterm infants who received probiotics. This reduction persisted up to 5 months of age and suggests a potential role for probiotics in limiting the development of antibiotic resistance [21].

To date, over 5000 preterm infants have received probiotics during randomized control trials [13], with an additional 21,000 infants included in cohort studies [22], not to mention the numerous preterm infants who have received probiotics outside clinical trials. Among all those infants, the incidence of adverse events has been extremely low. As previously mentioned, probiotics do not increase the risk of neurological impairment, neurosensorial complications or respiratory issues.

Despite the global reduction in LOS incidence, few cases have been reported of sepsis caused by probiotic strains or infection caused by contamination of the probiotic product. *Saccharomyces* spp. has been implicated in most sepsis cases [23,24], which highlights its unsuitability for use in preterm infants [25]. Three cases of *Bifidobacterium longum* bacteriemia were reported in Switzerland in 2015 among newborns receiving this probiotic strain. Two of them were asymptomatic and did not require treatment, while the third had necrotizing enterocolitis [26]. Another concern regarding probiotic use is the potential severity of sepsis resulting from contamination of probiotic product with pathogenic microorganisms [27]. This emphasizes the need for stringent quality control measures for probiotic products used in preterm infants.

Despite the substantial evidence supporting the administration of probiotics and the low occurrence of adverse events, there has been no definitive recommendation for their routine use until recently. The primary concern lies in the selection of probiotic strains due to the heterogeneity observed in the products tested in both randomized controlled trials and cohort studies. While safety appears to be recognized across all tested strains, variations in efficacy have been observed depending on the specific strains and combinations utilized [13].

## 2. Materials and Methods

### 2.1. Study Design

This study employed an observational, multicentric, transversal design to investigate medical practices related to probiotic use. The analysis focused on three categories: routine use, occasional use and absence of use.

### 2.2. Settings

The primary objective of the study was to determine the proportion of Neonatal Intensive Care Units (NICUs) in French-speaking European countries that routinely use probiotics. The secondary objectives included comparing administration protocols among NICUs and exploring the reasons for non-use.

The data were collected through an online questionnaire administered between October 2020 and June 2021, spanning a duration of 9 months.

The questionnaire was conducted using the Limesurvey website, and hosted on the server of the University of Caen Normandy. Responses were also stored through the Limesurvey website, ensuring anonymization of the physicians but enabling the identification of each response according to the represented NICU, to avoid duplication and facilitate data categorization by country.

Participants received clear information prior to their involvement in the study, and data were securely stored on the server of the University of Caen Normandy.

Approval for this study was obtained from the Local Health Research Ethics Committee of the Caen Normandy University Hospital under ID 2074.

### 2.3. Participants

The study involved European NICUs. For France, Switzerland and Belgium, we listed all the NICUs (67 in France, 9 in Switzerland, and 19 in Belgium). We obtained e-mail addresses through private contact lists, or hospital websites, and sent the questionnaire via email to at least one physician per unit. For the remaining European countries, the questionnaire was sent by e-mail to the respective national neonatal societies (or pediatric societies in the absence of a neonatal society), with a request for distribution to physicians working in NICUs.

The questionnaire link was sent via email to targeted physicians between October 2020 and June 2021, with a maximum of 5 reminders.

### 2.4. Variables

The primary endpoint of the study was to describe the use of probiotics at the NICU level, based on the answer to the second question: routine use, occasional use or absence of use. Secondary endpoints included comparing administration protocols, exploring reasons for non-use, and examining any correlation between probiotic use and the size of the NICU.

### 2.5. Data Sources/Measurements

The questionnaire was provided in French for France and French-speaking areas from Belgium and Switzerland, while other regions received the English version.

The first question of the survey collected information about the hospital’s name, city and country, to facilitate the tracking of responses and prevent duplication. The second question inquired about the type of probiotic use: routinely, as part of a research protocol, occasionally or never. Subsequent questions varied depending on the type of probiotic use, including inquiries about administration protocol or reasons for non-use, or both in the case of occasional use.

The second part of the questionnaire consisted of 12 common questions for all participants, covering aspects such as NICU size, typical patient profiles, and feeding protocols for preterm infants.

The questionnaire comprised open-ended, single-response and multiple-response questions. Only the first two questions were mandatory, the others were optional. Physicians had the opportunity to upload their administration protocol on the Limesurvey website.

The number of deliveries in 2019 at the hospital’s maternity ward was recorded for each French NICU using the scopesante.fr website, ensuring accurate data for comparing the NICUs.

Only one response per NICU was included, and if multiple responses were received from the same department, the most complete response was chosen based on the order of receipt.

### 2.6. Quantitative Variables

Data collected on the Limesurvey website were exported to Excel in spreadsheet format. Subgroup analysis were conducted for countries where more than 50% of answers were received.

### 2.7. Statistical Methods

For these countries, the frequency of each type of probiotic use was estimated using the Wald 95% confidence interval.

Subsequently, the responses were analyzed separately based on the type of probiotic use, with descriptive statistics, like percentages. For routine use, administration protocols were compared in terms of indications, contraindications, initiation time, and treatment duration. Indications and contraindications were compared for occasional use. For both types of use, the specific probiotic strains, pharmaceutical products, doses, and administration schedules were described.

Additionally, obstacles for routine use in the presence of occasional or no use were explored.

The pvalue.io (accessed on 20 April 2023) software was used to conduct univariate analysis using Chi2 and Fisher’s tests, aiming to identify a potential relationship between department characteristics and the type of probiotic use. This analysis was initially performed with three groups: routine use, occasional use, and absence of use. Subsequently, the occasional use and absence of use groups were combined into one category for further analysis.

## 3. Results

### 3.1. Participating Centers

All 67 NICUs in France, 9 NICUs in Switzerland, and 20 NICUs in Belgium were contacted through at least one of their physicians. Additionally, 31 European neonatal or pediatric societies were contacted, 5 societies agreed to forward the questionnaire, 2 refused, and the others did not respond. A total of 109 responses were received, of which 20 were excluded (Figure 1). Ultimately, 89 responses were retained, representing 11 different European countries (Figure 2).

A response rate of 85% was obtained for France, and 89% for Switzerland. Five responses initially classified as routine use were reclassified to occasional use because the administration was according to specific indications.

### 3.2. Frequency of Probiotic Use

From all the responses, 33% reported routine use, 22% reported occasional use and 45% reported an absence of probiotic use.

Among French NICUs, 53% reported never using probiotics (*n* = 30, IC95% 0.46–0.60), 21% reported routine use (*n* = 12, IC95% 0.16–0.26), and 26% reported occasional use (*n* = 15, IC95% 0.20–0.32). All eight responding Swiss NICUs reported routine probiotic use. Due to a low response rate from other countries, we were unable to evaluate the frequency of probiotic use.

Upon conducting univariate analysis, no significant differences were observed in type of probiotic use according to breastfeeding rates, characteristics of hospitalized newborns, or the annual number of births.

### 3.3. Comparison of Routine Use Protocols

#### 3.3.1. Indications

Regarding probiotic indication, a criterion based on gestational age was reported by 86% of NICUs, with different thresholds but a predominant indication in 41% of cases for infants with gestational age less than 32 weeks. A total of 55% of NICUs also use birth weight to define probiotics indications, with a predominant indication in 34% of cases for very low birth weight (VLBW) infants (under 1500 g).

#### 3.3.2. Contraindications

The primary contraindications for probiotics use were fasting (72%) and necrotizing enterocolitis (66%, Table 1).

#### 3.3.3. Initiation of Probiotics

Probiotics are administered in the first 48 h of live for 69% of NICUs, 72 h for 10% and later for 14%.

#### 3.3.4. Duration of Treatment

The duration of treatment is based on age in 34% of NICUs. Among them, 17% end the administration close to term (>36 weeks), 14% at an earlier stage (32–35 weeks), and 3% at 1 or 2 months of corrected age. In 17% of NICUs, the treatment continues until discharge home. In contrast, 38% of NICUs use a fixed duration of treatment: 7% for 10 days, 3% for 14 days, 14% for 28 days and 10% until the end of the bottle. Finally, 3% of NICUs discontinue the treatment when infants are completely enterally fed.

### 3.4. Comparison of Occasional Use Protocols

Indications for occasional use of probiotics include newborn colics in 55% of cases, antibiotic therapy in 40% of cases, and diarrhea or abdominal bloat in 35% of cases each. Other indications such as gastroeosophageal reflux, constipation, diaper rash or multiresistant bacterial colonization were reported in less than 10% of cases.

Contraindications for occasional probiotic use are not clearly defined, due to the lack of a specific protocol. However, many physicians rather like not to give probiotics to the most immature infants, with varying thresholds. Some physicians set a threshold of 1 kg of body weight, while others wait until corrected term. Additionally, two physicians reported the central veinous line as a contraindication.

### 3.5. Different Probiotic Strains Used

A total of 63% of Europeans NICUs use only a single probiotic strain, 20% use an association of two strains, and 10% use an association of three strains.

In France, probiotics used consist of a single strain of *Lactobacillus* spp., while in Switzerland, all NICUs use an association of *Bifidobacterium* spp. and *Lactobacillus* spp.

Among all the responses received, 63% of NICUs use only probiotics from the genus *Lactobacillus*, 29% use an association of strains from *Lactobacillus* and *Bifidobacterium*, and 2% use an association of *Bifidobacterium* and *Streptococcus*. A total of 15 different probiotic products have been reported, including 16 different bacterial strains (Table 2).

### 3.6. Rhythm of Administration

The rhythm of administration varies between one to four times a day, with differences observed between countries. In France, probiotics are mainly administered once a day (70% of NICUs), while in Switzerland, they are administered twice or four times a day (37.5% of NICUs each).

### 3.7. Daily Dose

Daily doses are usually between 0.1 to 1 billion CFU (Colony Forming Units) per day, for both routine and occasional use. However, higher doses have also been reported, reaching up to 30 billion CFU per day for occasional use (Figure 3).

### 3.8. Barriers to Routine Use of Probiotics

Among European NICUs that did not use probiotics routinely (occasional or absence of use), the most commonly reported obstacles are a lack of recommendations from endorsed societies (67%), lack of scientific evidence (38%), difficulty in obtaining probiotic products (21%), and fear of adverse effects (20%, Figure 4).

## 4. Discussion

This study reveals that the use of probiotics in European NICUs varies significantly. Among the countries included, Switzerland has a 100% rate of routine probiotic use, while France has a rate of 21%. The administration protocols and choice of probiotic strains differ widely among the NICUs. The main reasons cited for not using probiotics are the lack of recommendations and scientific evidence.

Comparisons with previous studies reveal significant variations in the rates of routine probiotic use across different regions. Probiotics are used by 8.8% of NICUs in United States [28], 17% in England [29], 19% in Germany [30], and 100% in New-Zealand [31]. In Canada in 2014–2015, 21% of preterm infants under 29 GW received prophylactic probiotics [32]. Viswanathan et al. in 2016 reported also a heterogeneous use of probiotics in United States, with an introduction mainly occurring at the initiation of enteral feeding, and a duration of treatment ranging from a few days until discharge home. A total of 16 different pharmaceutical products were used, only 4 of which have been studied for VLBW infants in randomized controlled trials [28]. Among European studies conducted between 2008 and 2018, most of them used *Lactobacillus rhamnosus*, with an association of at least two different strains in 53% of trials. In contrast, in this study, only 30% of European NICUs used an association of different strains, and none in France, and the genus *Lactobacillus* is the most frequently used in France. Administration duration and doses were found to be highly diverse across studies, showing a similar order of magnitude as observed in this study. However, none of the administration protocols have been able to demonstrate superiority over others in terms of efficacy [33].

In terms of strain selection, several meta-analyses have compared the administration of a single strain of *Lactobacillus* or *Bifidobacterium* with the use of multiple strains involving a combination of at least two genres. These studies have shown the superiority of the association of *Bifidobacterium* spp. and *Lactobacillus* spp. in reducing NEC and death [34,35]. Based on these findings, the European Society for Paediatric Gastroenterology, Hepatology, and Nutrition (ESPGHAN) has issued a position paper recommending specific strains for probiotic use. The paper recommends the use of either *Lactobacillus rhamnosus* GG ATCC 53103 or the combination of *Bifidobacterium infantis Bb-02*, *Bifidobacterium lactis* Bb-12 and *Streptococcus thermophilus* TH-4 [25,36,37].

Indeed, while the recommendations from the ESPGHAN position paper provide guidance on specific strains that have shown promise in preterm infants, they do not offer detailed instructions on the administration protocol for probiotics. Important considerations such as the initiation and duration of treatment are not fully addressed in these recommendations, leaving room for variations in clinical practice [25]. To note, the questionnaire was administered prior to the publication of the EPSGHAN position paper, which may have contributed to an increased response rate due to the absence of official recommendations at the time.

In 2020, the American Gastroenterological Association published recommendations about probiotic use in different gastrointestinal disorders, and recommended the use of an association of *Lactobacillus* spp. and *Bifidobacterium* spp. for preterm infants for the prevention of NEC [35,38]. However, a few months later, the American Academy of Pediatrics issued recommendations that did not support the use of probiotics in preterm infants. This decision was based on the lack of FDA-regulated pharmaceutical-grade probiotic products available in the United States [39].

The safety aspect of probiotic use in preterm infants is indeed a critical consideration. Most probiotic products worldwide are marketed as dietary supplements rather than medicines, and current legislation governing dietary supplements is less stringent. Scientific societies raised concerns regarding the quality of probiotic products, the quality control process and potential discrepancies between the label and actual content [40]. This lower level of regulation increases the risk of non-compliance with product standards. A composition analysis of 16 probiotic products revealed that only one of them matched its label claims perfectly, while the others exhibited both pill-to-pill and lot-to-lot variations [41]. Contamination of probiotic products can result in ingestion of unexpected pathogens, potentially leading to severe infections, like the one case of fatal gastrointestinal mucormycosis associated with a contaminated dietary supplement [27]. This highlights the need for careful monitoring of probiotic production, as is required for medicines.

Recent recommendations emphasize the importance of good quality control of the probiotic product chosen for administration to preterm infants, and parental information about the physician’s choice regarding probiotic administration to their infant. Another important point underlined by these recommendations is the ability of the microbiology laboratory to identify probiotic bacteria in blood cultures, in case of sepsis due to probiotic species [25].

Data about long-term outcomes are also limited to guide assessment of long-term efficacy. In a randomized trial of very low-birth-weight infants with follow-up of 249 infants at 18–24 months’ corrected age, the use of *Lactobacillus reuteri* did not increase or decrease the risk of adverse neurocognitive outcomes, assessed using the Bayley Scales of Infant and Toddler Development II [42].

According to recent studies, there is a consensus that future probiotic trials should focus on comparing different probiotic products against each other and comparing different administration protocols, rather than comparing probiotic administration to a placebo [13,43]. Simultaneous analysis of stool samples could provide valuable insights into the effects of probiotic administration on gut colonization and the persistence of probiotics in stool after discontinuation of treatment. This approach could help determine the optimal duration of treatment.

### Study Limitations

The strength of this study lies in the high response rates in France and Switzerland, which provide a reliable representation of probiotic use in these two countries. However, the study does have some limitations. Firstly, there is a potential selection bias due to the use of two different methods of contact. The method involving neonatal societies resulted in a low response rate for other European countries, which hindered the accurate estimation of probiotic use in those regions. Furthermore, the data collected in this study are based on self-reporting, which introduces the possibility of recall bias and potential misclassifications. It is important to acknowledge that participants may not accurately recall or report their probiotic usage, leading to potential inaccuracies in the data. Finally, it is also possible that physicians who actively use probiotics were more inclined to complete the questionnaire, introducing a non-response bias and potentially leading to an overestimation of probiotic use.

## 5. Conclusions

Despite the demonstrated benefits of probiotics in reducing NEC, sepsis and mortality in preterm infants, the rate of routine probiotic use remains low in European NICUs. Comparison of administration protocols reveals a great heterogeneity, reflecting the lack of comparability between studies on this subject. Considering that probiotics are intended for preventive use, ensuring their safety is of paramount importance for routine implementation. This study highlights the importance of rigorous microbiological quality control measures for probiotic product production to ensure their safety and efficacy in NICUs. Additional studies are needed to guide clinicians in choosing the most appropriate probiotic product to decrease NEC. Studies should focus on comparing different probiotic products and administration protocols, as well as exploring the impact of probiotic administration on gut colonization and the persistence of probiotics in the stool. These efforts will contribute to enhance the evidence-based practice of probiotic use in the care of preterm infants, ultimately improving their outcomes and reducing the risk of neonatal complications.

## Figures and Tables

**Figure 1 children-10-01889-f001:**
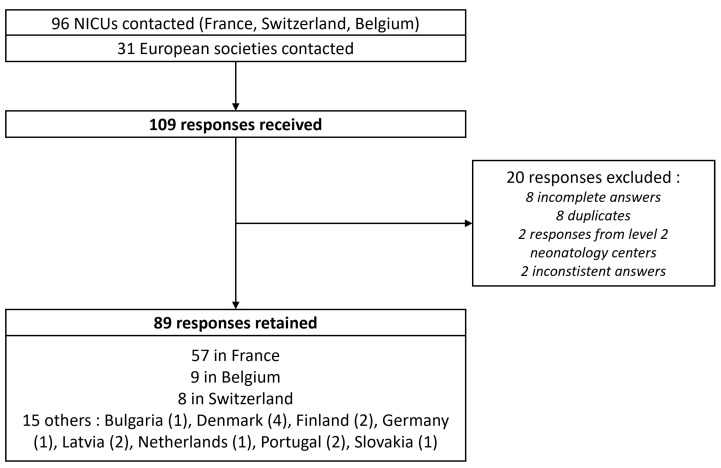
Flow chart of the study. NICU: Neonatal Intensive Care Unit.

**Figure 2 children-10-01889-f002:**
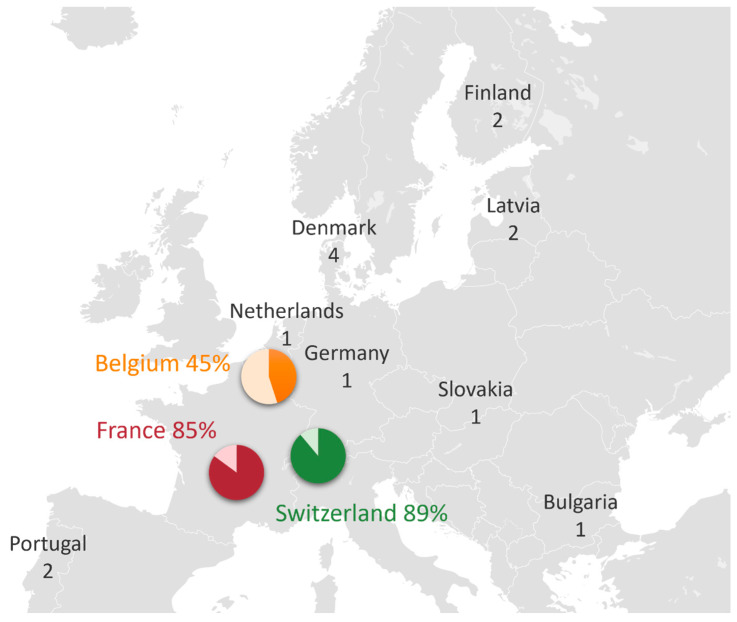
Geographical distribution of questionnaire responses among countries: each diagram represents the proportion of NICUs that responded to the questionnaire out of the total number of NICUs in each country. For other countries that were contacted through their societies, the number of responses is reported.

**Figure 3 children-10-01889-f003:**
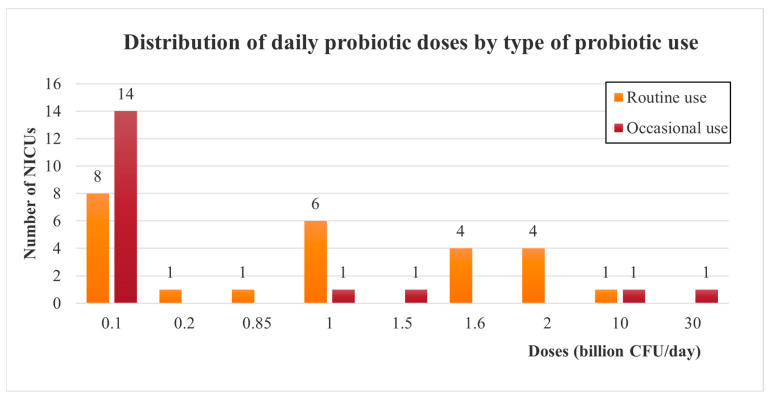
Doses of probiotics administered among European NICUs, depending on the type of probiotic use (routinely or occasionally). CFU: Colony Forming Unit.

**Figure 4 children-10-01889-f004:**
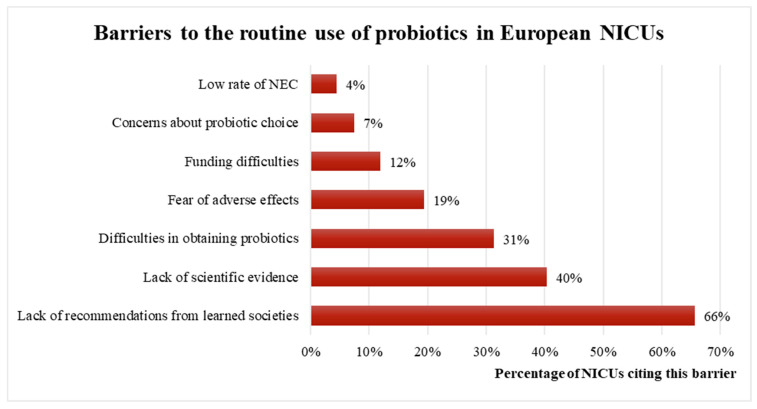
Barriers to the routine use of probiotics among European NICUs. NEC: necrotizing enterocolitis.

**Table 1 children-10-01889-t001:** Contraindication for probiotic routine use in European neonatal intensive care units (NICUs). HIV: Human Immunodeficiency Virus; NEC: Necrotizing Enterocolitis.

Possible Contraindication for Probiotic Use	Number of NICUs Using This Contraindication (*n*, %)
Fasting	21 (72%)
NEC	19 (66%)
Palliative care	9 (31%)
Sepsis	8 (28%)
Digestive malformation	7 (24%)
Maternal HIV Infection	2 (7%)
Antibiotherapy	2 (7%)

**Table 2 children-10-01889-t002:** List of probiotic products cited, with their composition and repartition of their routine or occasional use in European NICUs. N/A stands for unknown data about probiotic composition or probiotic commercial name. The compositions of the products were obtained from laboratories websites, with the last update conducted on 3 June 2023.

Commercial Name of Probiotic Product	Number of Strains	Bacterial Strains	Number of NICUs for Routine Use (*n*, %)	Number of NICU for Occasional Use (*n*, %)
Bifiform^®^	2	*Lactobacillus rhamnosus* LGG *Bifidobacterium lactis* BB-12	1 (3.4%)	
Biogaia^®^	1	*Lactobacillus reuteri* DSM 17938	7 (24.1%)	12 (60%)
Infloran^®^	2	*Bifidobacterium bifidum* NCDO 2203 *Lactobacillus acidophilus* NCDO 1748	4 (13.8%)	
Labinic^®^	3	*Lactobacillus acidophilus* NCFM *Bifidobacterium infantis* Bi-26 *Bifidobacterium bifidum* Bb-06	1 (3.4%)	
Lactéol^®^	1	Inactivated *Lactobacillus* LB	1 (3.4%)	2 (10%)
LCR restituo^®^	1	*Lactobacillus ramnosus* Lcr35 (Lcr Restituo)	1 (3.4%)	
Lenia^®^	1	*Lactobacillus rhamnosus var casei*	4 (13.8%)	
Liveo^®^	2	*Lactobacillus rhamnosus* LGG *Bifidobacterium lactis* BB-12		2 (10%)
Pharmalp Defense^®^	3	*Lactobacillus helveticus* R0052 *Bifidobacterium bifidum* R0071 *Bifidobacterium infantis* R0033	3 (10.3%)	
Probactiol Mini^®^	2	*Lactobacillus rhamnosus* LGG *Bifidobacterium lactis* BB-12	2 (6.9%)	
Progallia^®^	1	*Lactobacillus reuteri Protectis* DSM 17938		1 (5%)
ProPrems^®^	3	*Bifidobacterium infantis* Bb-02 DSM 33361 *Bifidobacterium lactis* BB-12^®^ *Streptococcus thermophilus* TH-4	1 (3.4%)	
Rela Drops^®^	1	*Lactobacillus reuteri*	1 (3.4%)	
N/A	2	*Lactobacillus rhamnosus* GG *Bifidobacterium lactis*		1 (5%)
N/A	1	*Lactobacillus reuteri*	1 (3.4%)	1 (5%)
N/A	N/A	N/A	2 (6.9%)	1 (5%)

## Data Availability

The data presented in this study are available in article.

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
