# Peer review of "State of the Art of Probiotic Use in Neonatal Intensive Care Units in French-Speaking European Countries"

_children, 2023, doi:10.3390/children10121889_

Round 1

Reviewer 1 Report

Comments and Suggestions for Authors

According to the paper by Amélie Blanchetière and colleagues, "State of the art of probiotics use in European Neonatal Intensive Care Units". Numerous studies have demonstrated the effectiveness of probiotics in reducing the incidence of necrotizing enterocolitis. However, the use of probiotics in preterm infants remains controversial. The purpose of this study is to evaluate the rate of probiotic use in European neonatal intensive care units (NICUs), to compare administration protocols, and to identify barriers and concerns associated with probiotic use in NICUs. Between October 2020 and June 2021, an online questionnaire was sent to all European NICUs via email. There were several questions related to the frequency of probiotic use. A study was conducted to collect information regarding probiotic administration protocols and reasons for non-utilization. There were 85% and 89% of responses from France and Switzerland, respectively. There is a routine use of probiotics in 21% of French NICUs and 100% of Swiss NICUs. There was significant heterogeneity in probiotic administration protocols, including variations in probiotic strains, administration methods, and treatment durations. The main obstacles to routine probiotic use were the lack of recommendations, a lack of consensus regarding strain selection, insufficient scientific evidence, and concerns regarding possible adverse effects. Routine probiotic administration remains low in European NICUs, with protocols differing widely. A further study is required to determine the most appropriate treatment modalities and to ensure the safety of the administration. Regarding the present manuscript, I would like to make a few comments.

There is no mention of Belgium in the abstract. Why is that?

Please change Figure 1 from French to English

This is a very good idea, however the introduction is divided into five chapters, perhaps a single paragraph would be more appropriate

It is possible that other variables should be included here, beyond the countries that participate, the duration of the probiotic treaty, and the frequency of its use

Is there a reason why the Swiss NICU uses 100 probiotics?

Thanks for the limitations section. Is it possible to enhance this survey, as the authors have other variables not included in the manuscript?

Author Response

Dear reviewer,

We sincerely appreciate the time and efforts you dedicated to reviewing our manuscript, along with the insightful remarks you provided.

Having carefully considered each of your concerns, we have provided detailed responses in the attached file. Your helpful feedback has been invaluable in enhancing the quality of our manuscript.

Thank you once again for your valuable contribution and commitment to improving our work.

Best regards,

Amélie Blanchetière

Reviewer 2 Report

Comments and Suggestions for Authors

General:

An interesting paper with data limited to French-speaking countries only, which makes its value rather local. Usage of the probiotics in other countries maybe totally different as result of the doctors’ habits and pressure from the industry. Therefore, the paper title should be modified to reflect this important factor.       

Detailed:

Line 31: genera or taxons but not types

Line 33: S.thermophilus is not a probiotic

Line 37: also medicinal products

Line 40: in neonates the most important is stimulation of the tight-junction proteins

Line 56: not true: coagulase-negative staphylococci in most European NICUs;

Line 123: French-speaking countries  

Line 137: how the questionnaires were distributed in general and in non-French speaking European countries? It looks that the distribution was very limited which makes quality of this survey rather limited.

Line 202: why Figure 2 is in French?

Line 222: how about Cesarean section?

Table 1: what is digestive malformation?

 Line 263: Enterococcus is not listed in Table 2.

Line 278: which probiotic was given in so high dose?

Author Response

(The authors gave the same response as above.)

Round 2

Reviewer 1 Report

Comments and Suggestions for Authors

Thank you for taking into account my previous comments. There was good argumentation in the answers and the information provided was well described. I have no further comments to make